# Bayesian learning of Causal Structure and Mechanisms with GFlowNets and Variational Bayes

## Abstract

Bayesian causal structure learning aims to learn a posterior distribution over directed acyclic graphs (DAGs), and the mechanisms that define the relationship between parent and child variables. By taking a Bayesian approach, it is possible to reason about the uncertainty of the causal model. The notion of modelling the uncertainty over models is particularly crucial for causal structure learning since the model could be unidentifiable when given only a finite amount of observational data. In this paper, we introduce a novel method to jointly learn the structure and mechanisms of the causal model using Variational Bayes, which we call Variational Bayes-DAG-GFlowNet (VBG). We extend the method of Bayesian causal structure learning using GFlowNets to learn not only the posterior distribution over the structure, but also the parameters of a linear Gaussian model. Our results on simulated and real-world data suggest that VBG is competitive against several baselines in modelling the posterior over DAGs and mechanisms, while offering several advantages over existing methods which include guaranteed acyclicity of graphs and unlimited sampling from the posterior once the model is trained.

## 1 Introduction

Bayesian networks (Pearl, 1988) represent the relationships between random variables as Directed Acyclic Graphs (DAGs). This modelling choice allows for inspecting of the conditional independence relations between random variables in a visual and straightforward manner. A Bayesian network can be used to represent a joint distribution over its variables. A *causal* Bayesian network, or a causal model, defines a family of distributions with shared parameters, corresponding to all possible interventions on the variables (Peters et al., 2017). Causal models allow for questions of importance to be answered such as: can we find an intervention that will result in the desired outcome in the modelled system? Answering these questions using causal models has been prominent in fields such as genetics (Belyaeva et al., 2021), medical diagnosis (Chowdhury et al., 2020) and economics (Awokuse, 2005).

The challenge of causal modelling lies in the fact that the search space of all possible DAGs grows super-exponentially in the number of nodes. As a result, many different heuristic methods have been suggested to tackle this problem. In addition to finding the graph, quantifying the uncertainty over causal models poses another challenge. Without assumptions on the DAG, observational data alone only identifies the Markov equivalence class of DAGs (Verma & Pearl, 1990), as a result, graph-finding algorithms that capture the uncertainty of the whole Markov equivalence class are beneficial, e.g., to avoid confidently wrong predictions or to explore in a way that will minimize this uncertainty. Reducing this uncertainty about the causal model requires interventions in the real world (Pearl, 2000) that could be prohibitively expensive, and quantifying uncertainty to direct the most informative interventions is a topic of particular interest in causal modelling.

Recently, a host of Bayesian causal structure learning algorithms that leverage the recent advances in gradient descent methods have been proposed (Lorch et al., 2021; Cundy et al., 2021; Deleu et al., 2022; Annadani et al., 2021). Each of these methods infers different aspects of the causal model and their uncertainty, given some assumptions on the model. The DAG-GFlowNet algorithm (Deleu et al., 2022) promises a unique way

of modelling a distribution over DAGs using GFlowNets (Bengio et al., 2021a;b). The method is limited to inferring only the DAG structure, without explicitly inferring the parameters of the causal mechanisms.

In this paper, we introduce an extension of DAG-GFlowNet which we call *Variational Bayes DAG-GFlowNet* (VBG), which infers the parameters of a linear Gaussian model between continuous random variables in the DAG, along with the graph itself, using Variational Bayes. This lends itself well to active learning approaches in causal modelling which require the mechanisms to be known (Scherrer et al., 2021; Agrawal et al., 2019; Tigas et al., 2022; Toth et al., 2022). Since conducting interventions to gather data for causal modelling is often very resource-heavy, active learning of causal structure is a promising direction of research. At this point, we would like to draw the attention of the reader to a concurrent work, JSP-GFN (Deleu et al., 2023) where the causal mechanisms are learned jointly with the causal graph using a GFlowNet, without any modelling assumptions on the mechanisms. We address the pitfalls of other Bayesian causal structure learning algorithms that model the mechanisms and the graphs. DiBS (Lorch et al., 2021) lacks the ability to sample from the posterior in an unbounded manner once the model is trained, and acyclicity is not guaranteed for the sampled graphs. Similarly, VCN (Annadani et al., 2021) does not guarantee acyclicity. BCD-Nets (Cundy et al., 2021) does not allow for flexibility of model parametrization for the mechanisms. Our method overcomes all these pitfalls and does comparatively well to these methods in our empirical evaluation using a number of metrics. In addition, we believe that this novel approach of finding the causal model using Variational Bayes has the potential to be applied to other Bayesian causal structure learning algorithms that only model the graph, and expand their capabilities to also model the mechanisms.

## 2 Related work

### 2.1 Causal structure learning

This paper contributes to the large body of work on causal structure learning, and relates most closely to *score-based methods* for causal discovery (Chickering, 2003). Score-based methods consist of two parts: defining a score that determines how well a DAG fits the observed data, and a search algorithm to search over the space of possible DAGs, to return the DAG with the highest score. This is in contrast to constraint based methods such as (Spirtes et al., 1993) that find the DAG that satisfies all of the conditional independence relations in the data. Examples of scores in score-based methods include the Bayesian Gaussian equivalent (BGe) score (Geiger & Heckerman, 2013) for linear Gaussian models, and the Bayesian Dirichlet equivalent (BDe) score (Chickering et al., 1995) for Dirichlet-multinomial models. The BGe score and the BDe score represent different ways to compute the marginal likelihood of the data, depending on the assumptions of the model. By using conjugate priors and likelihoods, the parameters of the mechanism can be marginalised out in closed-form. There are also a number of *hybrid* methods that combine score-based methods and constraint based methods such as (Tsamardinos et al., 2006) and more recently, GreedySP (Solus et al., 2021) and GRaSP (Lam et al., 2022). A subclass of score-based methods provides the exact identification of causal models from observational data using parametric assumptions (Hoyer et al., 2008; Shimizu et al., 2006). The modelling assumptions of Shimizu et al. (2006) show that if the mechanisms are linear and the noise is non-Gaussian, observational data is sufficient to identify the graph. Related to the work of Shimizu et al. (2006) in Peters & Bühlmann (2013), it was shown that linear Gaussian models with the same variance across nodes are identifiable using observational data alone. One of the more recent algorithms that resulted in a series of papers demonstrating its applications is the NO-TEARS framework (Zheng et al., 2018). In NO-TEARS, the score-based approach searching across discrete space with a hard constraint is relaxed into a soft constraint by using a differentiable function $h(W)$, which expresses the "DAG-ness" of a weighted adjacency graph $W$. The soft acyclicity prior and its variants lead to the development of many new causal structure learning algorithms using observational and interventional data, even with non-linear causal mechanisms (Yu et al., 2019; Lachapelle et al., 2019; Ke et al., 2019; Brouillard et al., 2020).

### 2.2 Bayesian causal structure learning

Bayesian approaches to structure learning involve learning a distribution over the graph and parameters of the mechanisms of causal models. However, computing the exact posterior from data requires calculating

the evidence, $P(\mathcal{D}) = \sum_G \int_\theta P(\mathcal{D} \mid G, \theta) P(G, \theta) d\theta$, $D$ being the data, $G$ the DAG and $\theta$ the parameters of the mechanism. This becomes quickly intractable as it involves enumerating over graphs and all possible parameters. As a result, variational inference or MCMC techniques are often used to obtain an approximate posterior over the graph and parameters. DiBS (Lorch et al., 2021) employs variational inference over a latent variable that conditions the graph distribution to approximate the posterior, in addition to the constraint proposed in (Zheng et al., 2018) as a soft prior, to perform marginal or joint Bayesian structure learning in non-linear or linear Gaussian models. VCN (Annadani et al., 2021) has a similar setup, but infers only the structure and attempts to capture a multimodal posterior by autoregressively predicting edges in the graph. However, due to the soft DAG-ness prior in both these methods, samples from the posterior approximation are not guaranteed to be valid DAGs. Similarly, Ke et al. (2022) define a distribution over graphs in an autoregressive way, but do not enforce acyclicity. BCD-Nets (Cundy et al., 2021) parametrize a distribution over DAGs as a distribution over upper-triangular weighted and permutation matrices, that once combined together induces a distribution over weighted adjacency matrices. This method is guaranteed to output only DAGs without the need for the soft DAG-ness constraint; however, it is only limited to linear Gaussian models. Subramanian et al. (2022) extends the BCD-Nets (Cundy et al., 2021) framework to learn a distribution over latent structural causal models from low-level data (e.g., images) under the assumption of known interventions. Wang et al. (2022) uses sum-product networks that define a topological ordering and then use variational inference to learn an approximate posterior. The structure MCMC algorithm (Madigan & York, 1995) build a Markov chain over DAGs by adding or removing edges. One of the motivations for learning a distribution over the causal model is that it lends well to active causal structure learning (Scherrer et al., 2021; Agrawal et al., 2019; Tigas et al., 2022; Toth et al., 2022). By knowing more about what the model is uncertain about, it becomes possible to choose the interventions that will result in the maximum reduction in the uncertainty of the causal model.

## 3 Background

### 3.1 Causal modelling

We study causal models described by a triple $(G, f, \sigma^2)$, where $G$ is the causal graph, $f$ represents the mechanisms between child-parent pairs, i.e., how a set of parent nodes influence a child node and $\sigma^2$ is the noise associated with each random variable in the graph. Assuming the Markov condition, the likelihood can therefore be factorized according to the following DAG formulation:

$$P(X \mid G, \theta) = \prod_{n=1}^{N} \prod_{k=1}^{K} P(X_k^{(n)} \mid \mathrm{Pa}_G(X_k^{(n)}), \theta), \tag{1}$$

where $K$ is the number of nodes in the graph, $N$ is the number of independent samples from the causal model, $\mathrm{Pa}_G(X_k)$ are the parents of $X_k$ in the graph $G$, and $\theta$ represents the parameters of the cause-effect mechanism, which we will call *mechanism parameters*. If we assume that the model has linear mechanisms with Gaussian noise (often called a linear Gaussian model), then the mechanism parameters $\theta$ correspond to a matrix of *edge weights* $\theta_{ik}$ representing the strength of the connection between a parent $X_i$ and a child $X_k$, through the following functional relation:

$$X_k = \sum_{i=1}^{K} \mathbb{1}(X_i \in \mathrm{Pa}_G(X_k)) \theta_{ik} X_i + \varepsilon_k$$
$$\text{with } \varepsilon_k \sim \mathcal{N}(0, \sigma_k^2). \tag{2}$$

It is well-known in causal modelling that a causal graph can in general only be identified up to its Markov equivalence class (MEC) given just observational data (Verma & Pearl, 1990). By introducing appropriate and enough interventional data, it is possible to narrow down the ambiguity to a single graph within a MEC. Given a dataset of observations $\mathcal{D}$, our goal in this paper is to model the joint posterior distribution $P(G, \theta \mid \mathcal{D})$ over graph structures $G$ and mechanism parameters $\theta$. Note that in this method, we assume the variances $\sigma^2$ are the same across all nodes.

## 3.2 Generative Flow Networks

Generative Flow Networks (GFlowNets; Bengio et al., 2021a) are generative models learning a policy capable of modelling a distribution over objects constructed through a sequence of steps. A GFlowNet has a training objective such that when the objective is globally minimized, it samples an object $x$ with probability proportional to a given reward function $R(x)$. The generative policy stochastically makes a sequence of transitions, each of which transforms a state $s$ (in an RL sense) into another $s'$, with probability $P_F(s' \mid s)$ starting at a single source state $s_0$, and such that the set of all possible states and transitions between them forms a directed acyclic graph. The sequence of steps to construct an object is called a "trajectory".

In addition to the structured state space, every transition $s \to s'$ in a GFlowNet is associated with *flow* functions $F_\phi(s \to s')$ and $F_\phi(s)$ on edges and states respectively, typically parametrized by neural networks. The GFlowNet is trained in order to satisfy the following flow-matching conditions for all states $s'$:

$$\sum_{s \in \mathrm{Pa}(s')} F_\phi(s \to s') = \sum_{s'' \in \mathrm{Ch}(s')} F_\phi(s' \to s'') + R(s'), \tag{3}$$

where $\mathrm{Pa}(s')$ and $\mathrm{Ch}(s')$ are respectively the parents and children of $s'$, and $R(s')$ is non-zero only at the end of a trajectory, when $s' = x$ is a fully-formed object. Intuitively, the LHS of equation 3 corresponds to the total incoming flow into $s'$, and the RHS corresponds to the total outgoing flow from $s'$. If these conditions are satisfied for all states $s'$, then an object $x$ can be sampled with probability $\propto R(x)$ by following the forward transition probability $P_F(s' \mid s) \propto F_\phi(s \to s')$ (Bengio et al., 2021a). It is worth noting that there exist other conditions equivalent to equation 3 that also yield similar guarantees for sampling proportionally to the reward function (Bengio et al., 2021b; Malkin et al., 2022; Madan et al., 2022). In particular, Deleu et al. (2022) used an alternative condition introduced by Bengio et al. (2021b) and inspired by the detailed-balance equations in the literature on Markov chains (see Section 3.3). In order to train the GFlowNet, it is possible to turn the target conditions such as the one in equation 3 into a corresponding loss function such that minimizing the loss makes the conditions satisfied, e.g.,

$$\mathcal{L}(\phi) = \mathbb{E}_\pi \left[ \left( \log \frac{\sum_{s \in \mathrm{Pa}(s')} F_\phi(s \to s')}{\sum_{s'' \in \mathrm{Ch}(s')} F_\phi(s' \to s'') + R(s')} \right)^2 \right], \tag{4}$$

where $\pi$ is some (full-support) distribution over the states of the GFlowNet.

## 3.3 GFlowNets for causal structure learning

Deleu et al. (2022) introduced a Bayesian structure learning algorithm based on GFlowNets, called *DAG-GFlowNet*, in order to approximate the (marginal) posterior over causal graphs $P(G \mid \mathcal{D})$. In this framework, a DAG is constructed by sequentially adding one edge at a time to the graph, starting from the fully disconnected graph over $K$ nodes, with a special action to indicate when the generation ends and the current graph is a sample from the posterior approximation. A transition $G \to G'$ here then corresponds to adding one edge to $G$ in order to obtain $G'$. Recall that a GFlowNet models a distribution proportional to the reward; therefore, DAG-GFlowNet uses $R(G) = P(\mathcal{D} \mid G)P(G)$ as the reward function in order to approximate the posterior distribution $P(G \mid \mathcal{D}) \propto R(G)$ once the GFlowNet is trained.

In order to train the GFlowNet, Deleu et al. (2022) used an alternative characterization of a flow network, different from the flow-matching conditions in equation 3. Instead of parametrizing the flows, they directly parametrized the forward transition probability $P_\phi(G' \mid G)$ with a neural network. Since all the states are valid samples from $P(G \mid \mathcal{D})$ (i.e., valid DAGs), they showed that for any transition $G \to G'$ in the GFlowNet, the detailed balance conditions (Bengio et al., 2021b) can be written as

$$\begin{aligned} R(G')P_B(G \mid G')P_\phi(s_f \mid G) = \\ R(G)P_\phi(G' \mid G)P_\phi(s_f \mid G'), \end{aligned} \tag{5}$$

where $P_B(G \mid G')$ is a fixed distribution over the parents of $G'$ in the GFlowNet, and $P_\phi(s_f \mid G)$ is the probability of selecting the terminating action to return $G$ as a sample from the posterior approximation.

Similar to Section 3.2, one can show that if the conditions in equation 5 are satisfied for every transition $G \to G'$, then the GFlowNet also induces a distribution over objects proportional to the reward (Bengio et al., 2021a); in other words here, DAG-GFlowNet models the posterior distribution $P(G \mid \mathcal{D})$. Moreover, we can also turn those conditions into a loss function to train the parameters $\phi$ of the neural network, similar to the loss function in equation 4.

Work concurrent to this paper is JSP-GFN (Deleu et al., 2023). JSP-GFN was an extension to the DAG-GFlowNet paper, where non-linear mechanisms are learned as well as the causal graph using the GFlowNet. As a result, the reward of the GFlowNet is a function of both the graph and the mechanism parameters. In practice, what this means is that the GFlowNet consists of a hierarchical model that transitions from state to state which are graphs, then given a graph, we transition to a particular value of the mechanisms parameters. This hierarchical transition function makes it possible to learn the transition within the space of graphs, then the space of parameters given the graph the benefit of JSP-GFN is that it is able to learn non-linear mechanisms between nodes. Our method presented in this paper, VBG takes a different approach of modelling the posterior distribution over the mechanisms and the graph, by having two separate modelling strategies for the mechanisms and the graph, and to update each iteratively. Although the drawback of VBG is that it is constrained only to learn linear mechanisms, in the experiments in this paper, where the data generation process is indeed linear, we demonstrated better performance of VBG in comparison to JSP-GFN for 20 and 50 node graphs.

## 4 Variational Bayes-DAG-GFlowNet

One of the main limitations of DAG-GFlowNet (Deleu et al., 2022) is that it only approximates the marginal posterior distribution over graphs $P(G \mid \mathcal{D})$, and requires explicit marginalization over the mechanism parameters $\theta$ to compute the marginal likelihood $P(\mathcal{D} \mid G)$, and as a result of the marginalisation, the mechanism parameters are never inferred. This limits the use of DAG-GFlowNet for downstream applications of causal structure learning such as inferring the range of possible causal effects of interventions or active intervention targetting.

In this work, we extend DAG-GFlowNet to model the posterior distribution $P(G, \theta \mid \mathcal{D})$. Allowing for the quantification of uncertainty of mechanisms as well graphs. To approximate the posterior distribution $P(G, \theta \mid \mathcal{D})$, we use Variational Bayes in conjunction with the GFlowNet. We model this posterior distribution using the following decomposition:

$$P(G, \theta \mid \mathcal{D}) \approx q_\phi(G) q_\lambda(\theta \mid G) = q_\phi(G) \prod_{k=1}^{K} q_\lambda(\theta_k \mid G), \tag{6}$$

where $\phi$ are the variational parameters of the distribution over graphs, $\lambda$ are the parameters of the distribution over mechanism parameters, and $\theta_k$ are the parameters of the mechanism corresponding to the variable $X_k$. Using this factorization, it is possible to use Variational Bayes to alternatively update the distribution over graphs $q_\phi(G)$, and the distribution over parameters $q_\lambda(\theta \mid G)$. We can write the Evidence Lower-Bound (ELBO) for this model as $\log P(\mathcal{D}) \geq \text{ELBO}(\phi, \lambda)$, where

$$\begin{aligned} \text{ELBO}(\phi, \lambda) = \mathbb{E}_{G \sim q_\phi} \big[ \mathbb{E}_{\theta \sim q_\lambda} [\log P(\mathcal{D} \mid \theta, G)] \\ - \text{KL}\big( q_\lambda(\theta \mid G) \,\|\, P(\theta \mid G) \big) \big] \\ - \text{KL}\big( q_\phi(G) \,\|\, P(G) \big). \end{aligned} \tag{7}$$

The derivation of the ELBO is available in A.1. Variational Bayes corresponds to coordinate ascent on $\text{ELBO}(\phi, \lambda)$, alternatively maximizing it with respect to the parameters $\phi$ of the distribution over graphs, and with respect to the parameters $\lambda$ of the distribution over mechanism parameters. Inspired by Deleu et al. (2022), we use a GFlowNet in order to model the distribution $q_\phi(G)$ (where $\phi$ are the parameters of the GFlowNet). We call our method *Variational Bayes-DAG-GFlowNet* (VBG).

### 4.1 Modelling the distribution over graphs with a GFlowNet

We show in A.2 that maximizing the ELBO with respect to the parameters $\phi$ is equivalent to finding a distribution $q_{\phi^\star}(G)$ such that, for any DAG $G$

$$\log q_{\phi^\star}(G) = \mathbb{E}_{\theta \sim q_\lambda}\big[\log P(\mathcal{D} \mid \theta, G)\big] \tag{8}$$
$$-\mathrm{KL}\big(q_\lambda(\theta \mid G) \,\|\, P(\theta \mid G)\big) + \log P(G) + \mathrm{cst}$$

where cst is a constant term independent of $G$. Equivalently, we can see that the optimal distribution $q_{\phi^\star}(G)$ is defined up to a normalizing constant: this is precisely a setting where GFlowNets can be applied. Unlike in Section 3.3 though, where the reward was given by $R(G) = P(\mathcal{D} \mid G)P(G)$, here we can train the parameters $\phi$ of a GFlowNet with the reward function $\widetilde{R}(G)$ defined as

$$\log \widetilde{R}(G) = \mathbb{E}_{\theta \sim q_\lambda}\big[w \log P(\mathcal{D} \mid \theta, G)\big]$$
$$- \mathrm{KL}\big(q_\lambda(\theta \mid G) \,\|\, P(\theta \mid G)\big) \tag{9}$$
$$+ \log P(G)$$

in order to find $q_{\phi^\star}(G)$ that maximizes equation 7. We introduce a weighting parameter that weights the likelihood term similar to what was done in Higgins et al. (2016). $w$ set to 0.1 which we found works well for all data sets that were tested in this paper. The weighting was found to be helpful to prevent the posterior collapsing to a point estimate.

In equation 9, the distribution $q_\lambda(\theta \mid G)$ corresponds to the current iteration of the distribution over mechanism parameters, found by maximizing the ELBO with respect to $\lambda$ at the previous iteration of coordinate ascent. Note that we can recover the same reward function as in DAG-GFlowNet by setting $q_\lambda(\theta \mid G) \equiv P(\theta \mid G)$. To find $\phi^\star$, we can then minimize the following loss functions, based on the conditions in equation 5

$$\mathcal{L}(\phi) = \mathbb{E}_\pi\left[\left(\log \frac{\widetilde{R}(G')P_B(G \mid G')P_\phi(s_f \mid G)}{\widetilde{R}(G)P_\phi(G' \mid G)P_\phi(s_f \mid G')}\right)^2\right], \tag{10}$$

where $\pi$ is a (full-support) distribution over transitions $G \to G'$. The same transformer architecture as used in Deleu et al. (2022) was used to parameterise both $P_\phi(s_f \mid G)$ and $P_\phi(G' \mid G)$. Furthermore, similar to Deleu et al. (2022), we can efficiently compute the difference in log-rewards, the delta score, $\log \widetilde{R}(G') - \log \widetilde{R}(G)$, necessary for the loss function in equation 10. The derivation of the delta score is given in A.3. Although applied in a different context, our work relates to the EB-GFN algorithm (Zhang et al., 2022) where an iterative procedure is used to update both a GFlowNet and the reward function (here, depending on $q_\lambda$), instead of having a fixed reward function.

### 4.2 Updating the distribution over mechanism parameters

Given the distribution over graphs $q_\phi(G)$, we want to find the parameters $\lambda^\star$ that maximize equation 7. To do so, a general recipe would be to apply gradient ascent over $\lambda$ in $\mathrm{ELBO}(\phi, \lambda)$, either for a few steps or until convergence (Hoffman et al., 2013). In some other cases though, we can obtain a closed form of the optimal distribution $q_{\lambda^\star}(\theta \mid G)$, such as in linear Gaussian models.

We present an example of the closed form update for a linear Gaussian model here for completeness. In the case of a linear Gaussian model, we can parametrize the mechanism parameters as a $K \times K$ matrix of edges weights, where $\theta_{ij}$ represents the strength of the connection between a parent $X_i$ and a child variable $X_j$; using this convention, $\theta_k$ in equation 6 corresponds to the $k$th column of this matrix of edge weights. We place a Gaussian prior $\theta_k \sim \mathcal{N}(\mu_0, \sigma_0^2 I_K)$ over the edge weights. The parameters $\lambda$ of the distribution over mechanism parameters correspond to a collection of $\{(\mu_k, \Sigma_k)\}_{k=1}^K$, where for a given causal graph $G$

$$q_\lambda(\theta_k \mid G) = \mathcal{N}(D_k \mu_k, D_k \Sigma_k D_k)$$
$$\text{with} \tag{11}$$
$$D_k = \mathrm{diag}\big(\{\mathbb{1}(X_i \in \mathrm{Pa}_G(X_k))\}_{i=1}^K\big).$$

In other words, the edge weights $\theta_k$ are parametrized by a Gaussian distribution and are then masked based on the parents of $X_k$ in $G$. Note that the diagonal matrix $D_k$ depends on the graph $G$, and therefore the covariance matrix $D_k \Sigma_k D_k$ is only positive semi-definite. We show in A.4 that given the distribution over graphs $q_\phi(G)$, the optimal parameters $\mu_k$ and $\Sigma_k$ maximizing equation 7 have the following closed-form:

$$\Sigma_k^{-1} = \frac{1}{\sigma_0^2} I_K + \frac{1}{\sigma^2} \mathbb{E}_{G \sim q_\phi} \left[ D_k X^T X D_k \right] \tag{12}$$

$$\mu_k = \Sigma_k \left[ \frac{1}{\sigma_0^2} \mu_0 + \frac{1}{\sigma^2} X_k^T X \mathbb{E}_{G \sim q_\phi} \left[ D_k \right] \right] \tag{13}$$

where $X$ is the $N \times K$ design matrix of the data set $\mathcal{D}$, $X_k$ is its $k$th column, and $\sigma^2$ is the variance the Gaussian likelihood model, and $\sigma_0^2$ is the variance of the prior of the mechanism parameters. We make an assumption that the variance of each node is equal, which is also an assumption made in Lorch et al. (2021), however this is not essential to the model, and the derivations can be adjusted to account for $K$ different variance parameters, one for each node. Note that while the true posterior distribution $P(\theta_k \mid G, \mathcal{D})$ is also a Normal distribution under a linear Gaussian model, we are still making a variational assumption here by having common parameters $\mu_k$ and $\Sigma_k$ across graphs, and samples $\theta_k$ are then masked using $D_k$.

We include the pseudo-code for VBG below:

---

**Algorithm 1:** Finding the posterior distribution of DAGs and linear parameters

---

**Data:** A dataset $\mathcal{D}$.
**Result:** A distribution over graphs $q_\phi(G)$ & a distribution $q_\lambda(\theta \mid G)$ over mechanism parameters.
**while** *number of iterations < max iterations* **do**
  **while** *delta loss of GFlowNet > min delta loss* **do**
  │  Update the GFlowNet using $\widetilde{R}(G)$ according to equation 9
  **end**
  Sample graphs from $q_\phi$, modelled with the GFlowNet
  Calculate closed form update of $\mu$ & $\Sigma$ according to equation 13 & equation 12, using sample graphs
**end**

---

### 4.3 Assumptions and limitations

A number of assumptions were made in order to model the posterior over DAGs and mechanism parameters. Firstly, we assume causal sufficiency; there are no unobserved confounders in the data. Secondly, we assume that the user has knowledge of the variance term in the likelihood model and it can be inputted in to the algorithm. In addition, our work is limited to cases where it is sufficient to model the mechanisms as multivariate Gaussians where the relation of parent-child nodes is linear. Finally, we assume that the linear mechanism parameters are sufficiently large, so that the strength of the connections between nodes is larger than the noise of the data, this is reflected in the experiments on synthetic data where the mechanism parameters are chosen to be outside of the range close to zero as outlined in section 5.1.

## 5 Experiments

We compare our method to other Bayesian approaches to causal structure learning that are able to learn a distribution over the graphs as well as infer the mechanisms between nodes. We examine the performance of methods by comparing the learned graphs to the ground truth graph, as well as the estimated posterior to the true posterior of graphs. Experiments were conducted on both synthetic Erdos-Renyi graphs and scale-free graphs (Erdös & Rényi, 1959) as well as real data from protein signalling networks (Sachs et al., 2005). Comparison of posteriors is only done for graphs with 5 nodes, where it is possible to enumerate all possible DAGs. We find that our method, VBG does comparably well to other methods across metrics. All results on scale-free graphs can be found on the appendix A.7.

## 5.1 Synthetic data generation

Experiments were conducted on $K = 5$ node graphs with 5 total edges in expectation, $K = 20$ node graphs with 40 total edges in expectation and $K = 50$ node graphs with 50 total edges in expectation. 20 different graphs from different seeds were generated according to the Erdos-Renyi random graph model as well as scale-free graphs (Erdös & Rényi, 1959). Results on scale-free graphs for 5 and 20 nodes can be found in the appendix A.7. 100 samples of data were taken from each graph. The mechanisms between parent and child nodes were linear Gaussian, with the edge weights sampled uniformly in $\{-2, -0.5\} \cup \{0.5, 2.0\}$ as done in Cundy et al. (2021). Given the ground truth graph, ancestral sampling was used with homogeneous variance across nodes of $\sigma^2 = 0.1$. Note that linear Gaussian models with homogeneous noise are provably identifiable using just observational data (Peters & Bühlmann, 2013) as a result, the uncertainty represented by the posterior in these results corresponds to uncertainty as a result of lack of data, not due multiple graphs being in the same Markov equivalence classes.

## 5.2 Baselines

We compare VBG to other methods that infer both the graph and the mechanisms. First, some recent gradient-based posterior inference methods, DiBS (Lorch et al., 2021) and BCD Nets (Cundy et al., 2021). In addition, we also compare against two MCMC graph finding methods Metropolis-Hastings MCMC and Metropolis-within-Gibbs with adjustments to find the linear Gaussian parameters as introduced in (Lorch et al., 2021), these are referred to as Gibbs and MH in the figures. We also compare to GES (Chickering, 2003) and the PC algorithm (Spirtes et al., 2000) using DAG bootstrap (Friedman et al., 1999) to obtain a distribution over graphs and parameters. Finally, we also compare VBG to other GFlowNet based methods, JSP-GFN (Deleu et al., 2023), which finds the causal mechanisms, as well as DAG-GFlowNet (Deleu et al., 2022). However since DAG-GFlowNet only finds the distribution over graphs and not mechanisms, we report the comparison of results in the appendix A.5.1.

For each metric, when calculating the expected value over graphs or parameters, $1,000$ samples each from the posterior over parameters and graphs are used. Results were calculated across 20 different graphs for the simulated data, and 20 different model seeds for the Sachs dataset. All box plots correspond to the median, 25th and 75th percentiles. For all the methods except BCD-Nets, the variance of the likelihood of the model for all nodes was set to the variance used to generate the data. For BCD-Nets the variance that is learned from the model is used to calculate the negative log-likelihood and the comparison to the true posterior. A uniform prior was used for all Bayesian methods, except for experiments with DiBS on graphs larger than 5 nodes, where an Erdos-Renyi prior is used. This is because DiBS requires sampling from the prior distribution of graphs for inference, and this is not possible with a uniform prior with more than 5 nodes.

## 5.3 Comparing the learned posterior to the ground truth posterior

We examine three features of graphs sampled from the posterior which we believe best summarises the distribution of graph structures. These are edge, path and Markov features as done in Friedman & Koller (2000), these can be seen in Figure 1. Edge features compare the existence of edges in graphs in the learned posterior and the true posterior, in summary it reflects the distribution of graphs at the most fine-grained scale compared to the other two metrics. Path features tell us about a longer-range relationship between nodes in the graph, looking at paths that exist between pairs of nodes through edges. Markov features compare the Markov blanket of each node in the graph, and tell us about conditional relationships between them. Note that these metrics only depend on the posterior over graphs and not parameters.

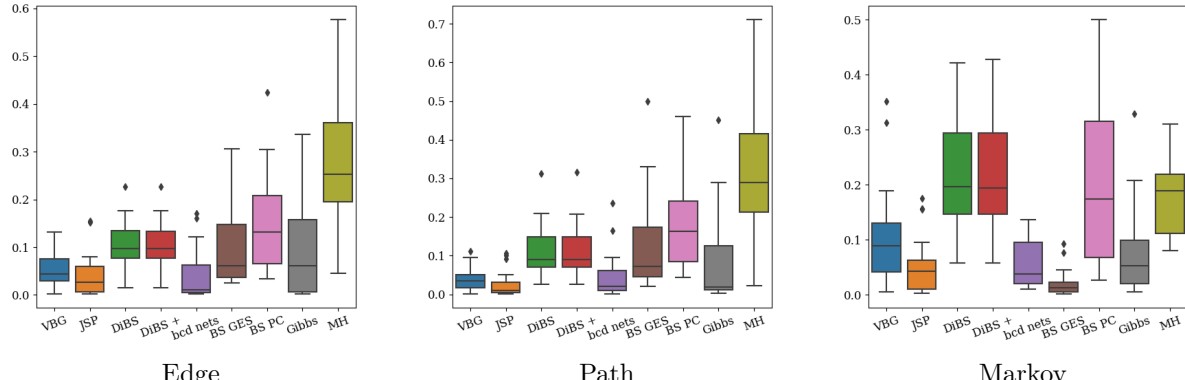

Figure 1: MSE of Edge, path and Markov features of the true posterior and the estimated posterior for 5 node Erdos-Renyi graphs (lower the better).

VBG does consistently well across all three metrics compared to baselines. However JSP-GFN and BCD-Nets outperform all other methods on these results including VBG.

## 5.4 Comparing the learned posterior to the ground truth graph

We look at metrics often quoted for pairwise comparison of graph structures to compare between the learned posterior and the ground truth graph for 5 node, 20 node and 50 node Erdos-Renyi graphs. Expected structural hamming distance ($\mathbb{E}$-SHD) represents the number of edge insertions deletions or flips in order to change the estimated graph to the ground truth graph, the lower the better, the results can be seen in Figure 2. Not this is the ($\mathbb{E}$-SHD) between estimated graphs and the ground truth DAG, not the CPDAG. Area under the ROC graph (AUROC; Husmeier, 2003) reflects the number of true positive edges and false positive edges predicted from the algorithm, the higher the better, which can be seen in Figure 3. These two metrics are more suited to maximum-likelihood estimates of graphs since they only compare against a single ground truth graph, but we include them nonetheless since they are also reported by other similar works. To assess the quality of the estimates of the mechanism parameters, we look at mean squared error of the mechanism parameters compared to the true graph in Figure 4. To assess both the quality of the learned mechanism parameters as well as the learned graph in we look at negative log-likelihood of held-out data given the learned posterior of graphs and mechanisms in Figure 5. Results for the Metropolis-Hastings algorithm are omitted from the figures as it did not perform as well as the other methods for mean squared error and negative log-likelihood, these results can be found in A.6.

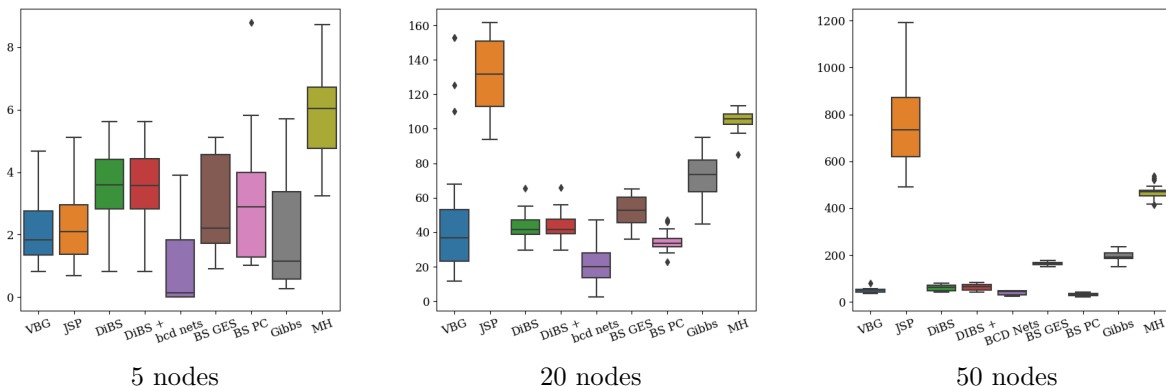

Figure 2: 𝔼-SHD (lower the better) inferring Erdos Renyi graphs with differing number of nodes. Box plots correspond to the median and 25th and 75th percentiles.

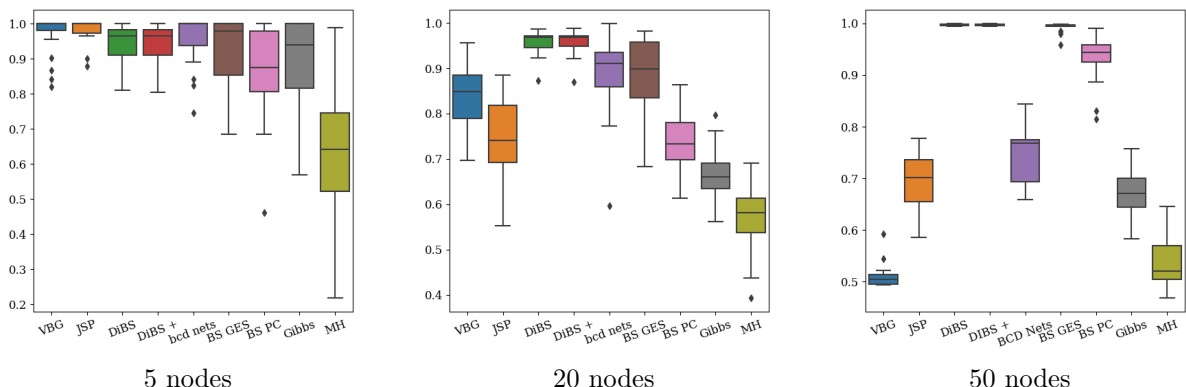

Figure 3: AUROC (higher the better) inferring Erdos Renyi graphs with differing number of nodes. Box plots correspond to the median and 25th and 75th percentiles.

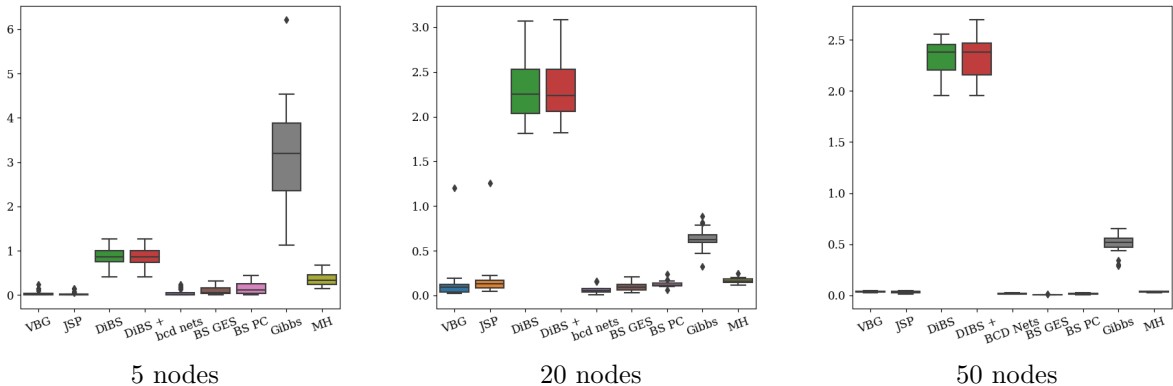

Figure 4: MSE $\theta$ (lower the better) inferring Erdos Renyi graphs with differing number of nodes. Box plots correspond to the median and 25th and 75th percentiles.

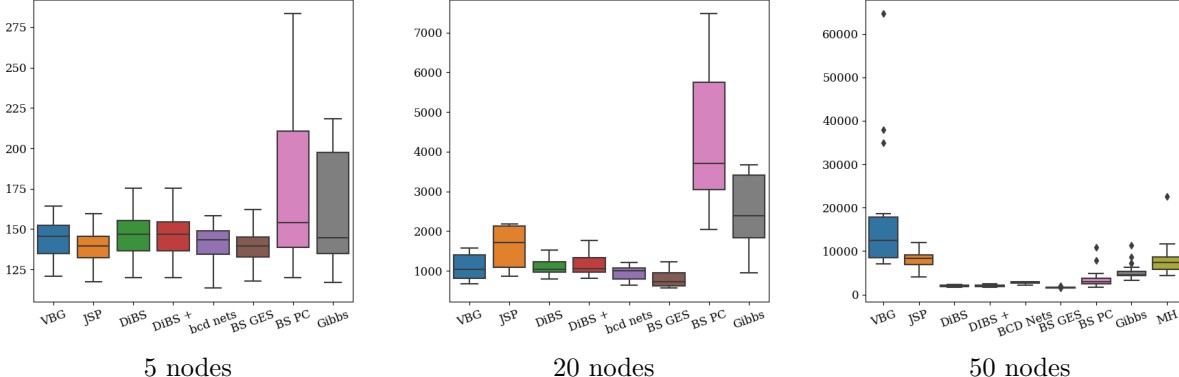

Figure 5: Negative log-likelihood of held-out data for Erdos-Renyi graphs with differing number of nodes (lower the better). Metropolis-Hasting (MH) results had extremely large values so were omitted here but can be seen in A.6. Results are averaged over 20 graphs

VBG performs well across metrics for 5-node graphs and 20-node graphs but does not perform as well compared to other benchmarks on 50-node graphs. VBG outperforms JSP-GFN across metrics for 20 node graphs. JSP-GFN performs well for 5 node graphs across all metrics but its performance is not maintained for 20 and 50 node graphs. Note that there is a difference between the graphs generated in the paper for JSP-GFN Deleu et al. (2023), where the mechanism parameters are sampled from a standard normal distribution, whereas in this work, the mechanisms parameters are sampled from $\{-2, -0.5\} \cup \{0.5, 2.0\}$ which more closes matches graphs generated in Cundy et al. (2021) and Lorch et al. (2021). We speculate that this resulted in larger magnitude of the raw values of the data, which lead to the instability of training JSP-GFN for larger graphs. BS-GES performs the best in terms of negative log-likelihood, achieving the lowest across all number of nodes.

Overall, VBG is competitive against baseline methods across these metrics but does not outperform baseline methods. We speculate that approximations which were necessary to be made in the development of the algorithm lead to biased estimates of the posteriors for VBG. The approximate posterior over parameters is learned jointly for all graphs sampled from the posterior, and should in practice, be learned for each graph. As a result, the calculation of the reward function, which is used to train the GFlowNet, which depends on the mechanisms parameters where also biased. Leading to biased estimates of the posteriors for both the graphs and the mechanisms.

### 5.5 Experiments on protein-signalling dataset

We performed experiments on the protein-signalling dataset (Sachs et al., 2005) consisting of observations of 11 proteins. We use a subset of this dataset consisting of 854 samples which are available on the bnlearn R-package (Scutari & Denis, 2021). AUROC curve and $\mathbb{E}$-SHD for the ground truth graph and the inferred graphs can be seen in Figure 6. VBG is competitive against baselines but does not outperform them.

## 6 Conclusion

We propose a method to model the posterior distribution over DAGs and linear Gaussian mechanisms between random variables using GFlowNets and Variational Bayes we call Variational Bayes DAG-GFlowNet (VBG). We found that the approximate posterior inferred using VBG is able to model well the true posterior when inspecting 5-node graphs according to the metrics we inspected. VBG is able to infer the ground truth graph for 5 node and 20 node graphs with mixed results for 50 node graphs compared to other baseline methods.

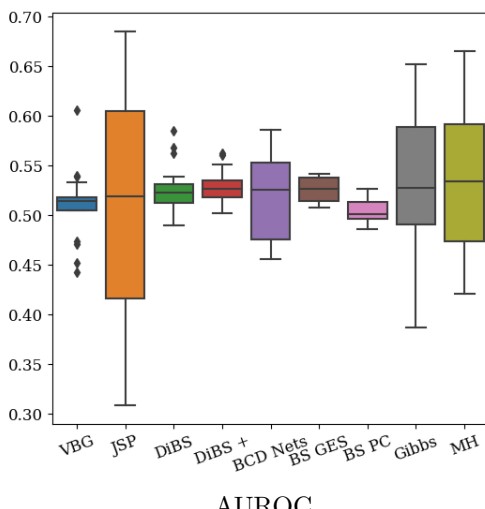 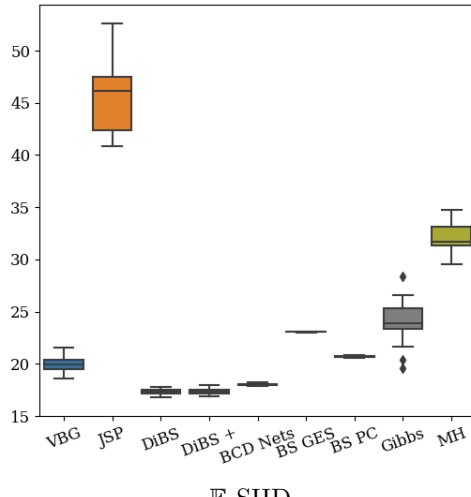

AUROC                                          𝔼-SHD

Figure 6: Sachs dataset. Seeds are over 20 model initialisations. Results also containing DAG-GFlowNet can be found in the appendix A.5.2

We believe that VBG offers something new to the toolbox for Bayesian causal structure learning in several ways. First, since VBG returns DAGs at every step of the training iteration, it lends itself well to using Variational Bayes to learn aspects of the causal model and the graph in this alternating step-wise procedure. Other existing Bayesian causal structure learning algorithms (Lorch et al., 2021; Annadani et al., 2021) rely on a soft acyclicity prior (Zheng et al., 2018) that does not guarantee sampling of DAGs, and often return a large proportion of cyclic graphs at the start of training. This may make these methods not amenable to iterative modelling and updating of graphs and other aspects of the causal model. For example the VBG setup could be extended to not just learning aspects of the causal model, but also to more fine-grained aspects of causal modeling such as inferring latent variables, or conversely for higher-level objects which are functions of the causal model as was suggested in Toth et al. (2022). Once VBG is trained, there is no upper limit on the number of samples that can be sampled from the posterior. This is in contrast to DiBS (Lorch et al., 2021), where the number of samples of the posterior must be pre-specified before training the model. In this paper, a linear Gaussian mechanism is assumed, however the assumption over the mechanisms is flexible for VBG as long as the delta score can be calculated from the likelihood of the model. This is in contrast to BCD-Nets which relies on a Gaussian linear mechanisms. Although JSP-GFN (Deleu et al., 2023) overcomes this by using neural networks to parameterise mechanism and therefore is not bound to this assumption, we show empirically that JSP-GFN fails to successfully infer the distribution over larger graphs using our simulated Erdos-Renyi graphs.

Going forward, we would like to do active intervention targeting using the uncertainty quantification of VBG (Scherrer et al., 2021; Agrawal et al., 2019; Tigas et al., 2022; Toth et al., 2022).

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
