# OpenReview forum: "Bayesian learning of Causal Structure and Mechanisms with GFlowNets and Variational Bayes"
_TMLR — Rejected by TMLR_

### Review · Reviewer_ZYGC · 2024-03-18

**Summary Of Contributions:**

As the authors put it: "In this work, we extend DAG-GFlowNet to model
the posterior distribution P(G,\theta|D)". This posterior is
approximated as in equation (6) and a "coordinate ascent" method is
used to alternate between updating the variational parameters for G
and \theta.

The paper is admirably succinct and the method employed is clearly
described. There is due discussion of the similarities and differences
from related methods (although at least one comparator is missing -
see below).

I have concerns, howeer, regarding the experimental methods and the results.

**Audience:**

Yes

**Claims And Evidence:**

Yes

**Requested Changes:**

1. Comparison with BiDAG
2. Experiments with varying size of dataset
3. Evidence of reasonable performance on larger number of nodes

TYPOS, etc

current iterate -> current iteration

them comparison -> the comparison

"informs of the number": this is not correct English.

**Strengths And Weaknesses:**

EXPERIMENTAL METHOD

This:
"We make an assumption that the variance of each node is equal, which
is also an assumption made in Lorch et al. (2021),"
seems like quite a strong assumption. Why make it? Someone else
assuming it is not sufficient justification.

Also why this:
"For all the methods except BCD-Nets, the variance of the likelihood of
the model for all nodes was set to the variance used to generate the
data."

In 5.1 we are told how graph structure and parameters are generated
but not how many synthetic datapoints are sampled using ancestral
sampling. Only with the Sachs dataset do we get told how many
datpoints (854) there are. We should not only be told the number of
datapoints, there should be *multiple experiments with different sized
synthetic datasets*. This would enable us to see which methods cope
well with pointy versus flattish posteriors.

EMPIRICAL RESULTS

I've seen good results from the BiDAG system of Kuipers et al, so I
think this should be one of the sytems compared to. BiDAG is included
in the benchpress system (
https://benchpressdocs.readthedocs.io/en/latest/ ), which might make
running comparisions easier.

It is useful to present comparative results using 5 node graphs, since
this can potentially provide insight into the competing
methods. However, since one can (as noted by the authors) enumerate
all 29281 DAGs with 5 nodes (and thus compute ground truth posterior
quantities), in any real application we care about performance with
a larger number of nodes. So good performance of VGB (or any other
method) on 5 nodes is not a strong reason to use VGB.

Section 5.3: VBG does respectably, however JSP-GFN and BCD-Nets do better.

Section 5.4: Although, as mentioned, the results in Figs 2 and 3
compare estimated posterior quantities to a fixed single graph, they
are still of interest. (If the data is large the posterior should be
concentrated around the ground truth, but we are not told the size of
the synthetic data). On 50 nodes - and it is the larger cases that
matter most, VBG does really rather poorly.

It would be interesting to see results on synthetic data which are not
restricted to linear Gaussian, although I would not go as far as to
make this a condition of publication.

CONCLUSION

As the authors argue the proposed method does have a number of
"selling points" and is an interesting approach which merits further
development. However, I just think publication of the current paper
would be premature. Empirical results, although often reasonable, are
sometimes poor, particularly for 50 nodes (Fig 7 in supp material is
particularly striking). Also further analysis - not just speculations
- helping us understand why we have these empirical results is needed.

---

> ### Author Response · Authors · 2024-04-26
> **Response to reviewer ZYGC**
>
> * **The variance of the model.** The method does not rely on the variances being equal, however, the method does rely on the variance of the linear Gaussian mechanisms to be input to the algorithm. We will make this clearer in the paper. This is the case with all of the algorithms to which we compared VBG, except for BCD-Nets. BCD-Nets can learn the variances. If the variance of each parent-child pair is different, we could include this in the modelling assumption of any of the algorithms in the experimental section and specify the different variances in the algorithm.
> * **Number of samples of data.** We removed the information about the number of samples by mistake, thank you for catching this, and we will include them in the updated paper version. We used 100 samples for the Erdos-Renyi experiments. We do not perform experiments with different number of samples, but we do provide experiments for different number of nodes. Since more samples of data are required for larger number of nodes, we believe that this is sufficient variation in the experimental setup.
> * **Comparison to the true posterior in 5 node graphs.** We can only compare the estimated posterior to ground truth posterior for 5 node graphs. The purpose of this is to show that VBG can be shown to model the posterior well. There is no way to compare against the true posterior for larger graphs, as a result, we are limited to showing its capabilities on 5 node graphs only.
> * **50 node graphs.** We are currently running experiments on 50 node graphs with reduced sparsity.
> * **Miss specified models.** We test our method on the Sachs dataset, which is not synthetic, as a result, we believe this is a good experimental setup for testing on data where the data-generating mechanism is not known to be linear Gaussian.

---

### Review · Reviewer_QuQY · 2024-03-27

**Summary Of Contributions:**

The paper studies the problem of Bayesian causal structure learning from observational data. The authors propose a method that uses variational inference and GFlowNets to learn and parameterize the joint posterior distribution over DAGs and mechanisms. Key modeling assumptions are that mechanisms are assumed to be linear, noise variances are equal, and there are no latent confounders. Some experiments are provided to demonstrate the usability of the method.

**Audience:**

Yes

**Broader Impact Concerns:**

No concerns.

**Claims And Evidence:**

No

**Requested Changes:**

Please see the section above on weaknesses. I think my main criticism of this work is that I could not find a clear reason on why this approach is "useful". Mainly based on the experiments that do not show a clear pattern on when or why to use this method. Also, it is important to report runtime since the whole point of causal discovery is to produce efficient methods that could reveal insights on cause-effect relation when dealing with large number of variables.

**Strengths And Weaknesses:**

Strengths
-----------

* The paper reads well in general modulo some minor details that would be good to incorporate.
* The problem setting is relevant in causality, although assuming Gaussian linear models and no latent confounders certainly hinders the significance.


Weaknesses
--------------

* A major weakness in my opinion is that I found the experiments rather unconvincing.
  1. Other types of random graphs should be experimented on. An important class is scale-free graphs, since the existence of hubs imposes a key statistical challenge for causal discovery methods.
  2. The sparsity level on 50 nodes is "too much", the number of expected edges is about  4% of the total number of possible edges. And at this level, the method seems to underperform. Figure 2 can be misleading for 50 nodes due to the scaling of the y-axis.
  3. Unless I missed it, the number of samples is not reported. I am curious on this since recent non-Bayesian methods such as [1, 2] (not discussed in the paper), have been shown to perform well even for denser graphs for this class of SEMs using around 1K samples. I think these methods could also serve as baselines beyond PC/GES.
  4. I do not get what's the point of testing on homoscedastic models if they are known to be identifiable, see also [3]. Cannot statistical errors be handled in a more efficient way than a fully Bayesian approach? I think, for non-identifiable models, a Bayesian approach makes more sense as it can shed light on potential target interventions.
  5. It is also important to run the method on misspecified models. In prior work it is common to generate data from LinGAMs even though Gaussianity might have been assumed.

* Something that concerns me is the main takeaway of the paper. In its current form it looks like another Bayesian approach without making sense of when or why I should use this approach. Sometimes having strong empirical results help, but this is not the case. Moreover, having JSP-GFN as a more flexible model for nonlinear mechanism makes one wonder what is the point of VBG. Perhaps a better tuning to JSP-GFN might make it work better for linear models as well?

* I am not sure about the criticism to other prior work such as DiBS or VCN about not having guaranteed acyclicity. They use the continuous formulation of DAGs that was shown to lack power to detect large cycles, see [2]. Maybe these approaches might improve their performance simply by replacing the type of continuous DAG formulation.

* A few informal statements are made in the paper that would be good to clarify or expand onto:
  1. In page 4: "...one can show that if the conditions in equation 5 are satisfied...", is this somewhere in the paper? I am not an expert in GFlowNets so I would appreciate more clarity on this.
  2. In Section 4.3: (i) "...there is some knowledge of the variance term in the likelihood model that the user selects for training the model...", could you elaborate what's 'some knowledge' here?  (ii) "...Finally, we assume that the linear mechanism parameters are sufficiently large...", this was also not clear at all in the experiments.

[1] Deng et al., ICML 2023. Optimizing NOTEARS objectives via topological swaps.\
[2] Bello et al. NeurIPS 2022. DAGMA: Learning DAGs via M-matrices and a Log-Determinant Acyclicity Characterization.\
[3] Loh & Buhlmann, JMLR 2014. High-Dimensional Learning of Linear Causal Networks via Inverse Covariance Estimation\

---

> ### Author Response · Authors · 2024-04-26
> **Responses to reviewer QuQY**
>
> * **Experiments on different graphs.** We are currently running experiments on scale-free graphs as, well as denser 50-node graphs. We will update the paper before the end of the rebuttal period when these analyses are completed.
> * **Number of samples.**  We removed the information about the number of samples by mistake (thanks for catching the mistake!), and we will include them in the updated paper version. We used 100 samples for the Erdos-Renyi experiments, and 854 samples for the Sachs dataset.
> * **Comparison to other methods.** We focused our analysis of comparisons against other Bayesian causal structure learning methods. We included two very standard non-Bayesian methods since these traditional methods are often compared against new causal structure learning methods rather than choosing some of the most recent non-Bayesian causal structure learning methods.
> * **Variance of model.** VBG, like all the methods that were compared to, requires that the variance of the linear Gaussian models be inputted into the algorithm. We will make this aspect clearer in the paper. If we are working with hetero-scedastic models, we would input all the different noise variances into the algorithm, for each parent-child pair.
> * **Miss-specified models.**  We do not perform experiments on miss-specified models. However, we test our method on the Sachs dataset, which is not data generated synthetically, so we believe this, in part, satisfies the need to explore the method on data that is known to be generated by a linear Gaussian mechanism.
> * **Usefulness of method.** VBG was not shown to beat other state-of-the-art methods; however, we still think that this paper is of interest to the causal structure learning community and the GFlowNet community. It demonstrates the ability of GFlowNets to be used for causal structure learning, using a variational Bayes approach.
> * **Criticism of other methods.** Methods that use the continuous DAG formulation suffer from the drawback of returning cyclic DAGs. These methods could replace this acyclicity prior, but this is not something that has been explored or demonstrated in these papers.
> * **Clarity of statements.** Thank you for pointing out the parts of the papers which are unclear, we will make sure that these parts are clarified in the text before the end of the rebuttal period.

---

### Review · Reviewer_RG1S · 2024-04-19

**Summary Of Contributions:**

The paper proposes a new Bayesian method for jointly learning a posterior over graphs and parameters for linear Gaussian DAG models with equal variances.

**Audience:**

Yes

**Claims And Evidence:**

No

**Requested Changes:**

*Critical*:
- It's stated on page 1 that DAGs can only be learned up to an MEC from observational data, but this is not true. Indeed, the authors (correctly) contradict that claim in Section 2.1 when mentioning the work on exact identification Ammend the first statement to something more like the second. A similar claim in made again on page 3 (but at least this time it says "in general", so it's not a false claim; also consider whether this repeition really necessary.)
- In Section 2.1, more recent, state of the art score-based methods like GreedySP and GRaSP aren't mentioned; MMHC is cited as score-based even though it's a hybrid method. Also critically neglecting MCMC methods, like order-based MCMC of Kuipers, Suter, Moffa (2022). The baselines against which VBG are compared are not convincing, since they neglect modern MCMC methods with good performance.
 - It looks in equation 2 like there are equal variances, however here it is presented more as a standard background, and only later (in a somewhat random, hidden spot on page 7) is it clarified that this work makes this strong assumption.
- Add more details to pseudocode in Algorithm 1. (I think including explicit pseudocode like this is great! It just needs a bit more detail.)
- Run experiments on more varied graphs, not just rather sparse ones (50 node graphs with expected degree of 1 is too limited); run experiments where the error variances are learned by all methods instead of giving this important ground truth information; compare to more standard state of the art causal discovery methods (see [this comparison](https://benchpressdocs.readthedocs.io/en/latest/examples.html#random-gaussian-sem) for example, especially omcmc and boss or grasp.
- mention runtime of VBG compared to other methods
- show AUROC curves in Fig 6, not a box plot
- be more careful about SHD comparisons; are the SHDs shown between DAGs or between CPDAGs (of MECs)?


*Suggested*:
- multple claims that interventions can be prohibitively expensive, and hence active learning via VBG is a contribution; can the authors elaborate/provide references about prohibitively expensive interventions?
- multiple mentions that a benefit of VBG is that it allows unbounded sampling from the posterior; this strikes me as a odd, considering in general learning a posterior allows unbounded sampling from it (though the authors do explicitly cite/explain that DiBS lacks this)
- I found the organization at the end of Section 1 through Section 2 a bit confusing; discussion of the related Bayesian gradient-based methods are interspersed with other discussion in a way that felt a bit like jumping around.
- inconsistencies about whether the object of the sentence is treated as singular or plural, especially at the end of Section 4. (e.g., "Similarly, Ke et al. (2022) define..." vs "Wang et al. (2022) uses...")
- In Section 3.1 "the mechanisms between child-parent pairs" seems different to me from "how a set of parent nodes influence a child node". For example, normality of marginals f(a) and f(b) does not imply normality of the joint f(a, b).
- write out "with respect to" or at least use "wrt" or "w.r.t." instead of "wrt."
- go carefully through paper to fix run-on sentences
- don't overload the X and X_k notation as in equations 12 and 13
- always capitalize names like Markov and Sachs

**Strengths And Weaknesses:**

*Strengths:*
- powerful, flexible, extensible framework (though these claims need some more support to be convincing, at least for someone like me with less familiarity with GFlowNets)

*Weaknesses:*
- some missing causal discovery literature/references, and false claims that follow from neglecting this literature
- restrictive model assumptions (equal variances, linear Gaussian), no (or not clearly stated) theoretical guarantees, and insufficient experimental results
- some typos, repeated sentences, etc. that make the paper harder to understand

---

> ### Author Response · Authors · 2024-04-26
> **Response to reviewer RG1S**
>
> * **Conflicting statements about MEC**. Thank you for pointing out some inconsistencies in our writing that we overlooked, we will be sure to change these when we submit our amendments to our paper.
> * **Comparison against more methods**. We were guided by other recent Bayesian causal structure learning work that appeared at ML conferences to decide which methods to compare our method against eg (JSP, DAG-GFN, BCD-Nets, DiBS). We understand that there are other methods we could have benchmarked in our paper, but we focused on Bayesian causal structure learning methods, and included some of the state-of-the-art methods  and regrettably missed others. We think that comparing against all state-of-the-art methods would be necessary to support a claim that VBG outperforms all recent Bayesian causal structure learning methods, however, we did not make such a claim. VBG does not outperform state-of-the-art methods which were compared against in the paper.  Secondly, we focused on comparing against other methods which were specifically Bayesian causal structure learning methods. However we did include two widely studied non-Bayesian methods which are often included as benchmarks in non-Bayesian causal structure learning papers. An extensive comparison against all causal structure learning methods, Bayesian or non-Bayesian we believe, would not add much to support the claims made in the paper.
> * **Learning the variance of model.** The methods we compared against do not have the capacity to learn these variances, except for BCD-Nets, which was run so that the variance is learned. All methods require the estimate for the variance to be inputted into the algorithm. We could design a way to estimate this variance, then input it into each algorithm, but we think that this would significantly change the method and the algorithm, so we will not include this in this rebuttal.
> * **Experiments on more graphs.**  We agree that the sparseness of 50 node graphs is indeed too much, we are currently in the process of running experiments with more edges. We also aim to include results for scale-free graphs before the end of the rebuttal period.
> * **Run time of methods**. We are currently performing an analysis on the run-time of VBG in comparison to other methods.

---

### Author Response · Authors · 2024-05-03
**Overall comment at the end of the rebuttal period**

We thank the reviewers for taking the time to read our paper. During the rebuttal period, we made corrections to the main text as suggested by the reviewers, as well as adding more experimental results to the paper.
We found that VBG is not able to infer graphs with 50 nodes, as was evident from the experimental results in the paper. We tried increasing the number of edges in the experiments from 50 to 200 but we found that VBG still had higher shd than other methods, and performs similarly to the results of the sparser graphs of 50 nodes and 50 edges in the experiments section.
However we report new results for scale-free graphs with 5 nodes and 20 nodes, and VBG performs well compared to other baseline methods. These results are included in the supplementary material.
At this point, VBG has not been optimised to be efficient, and we did not have the time to do this optimisation of the algorithm. We also did not have the time to optimise the methods that were bench-marked against (finding the minimum number of iterations to run as well as optimising the code), and so we thought that a comparison of the different methods at this point would not reflect the true run times of the algorithms so we have not included this in the paper.

---

### Decision · Action_Editor_Vg9x · 2024-05-20

**Recommendation:** Reject

**Comment:**

There were some consistent threads among the reviewers during discussion.
- The paper lacks any theoretical guarantees, even with strong assumptions
- There are missing comparisons both empirically and otherwise to established Bayesian causal literature. Among others, work by Kuipers et al. was mentioned independently by multiple reviewers.
- The comparisons to competing methods were underwhelming, and the paper did not provide a clear indication of when one would want to use VBG.

Aside from these, the reviewers noted various points raised during the discussion that they found were not adequately addressed in the rebuttal. Reviewer RG1S specifically mentioned their 2nd, 5th, 6th, and 8th points from their original review.

**Audience:**

The paper presents a method for Bayesian learning of causal structures, a topic that is of interest to the general TMLR community. However, reviewers consistently pointed out limitations in the experimental results, especially with regards to comparisons with existing methods, that indicate the method itself may not be of particular interest to the community without further development.

**Claims And Evidence:**

The main claim, which appears in the abstract, is that the new method (VBG) is competitive against baselines, and offers advantages over existing work such as acyclicity of graphs and unlimited sampling. The reviewers found the claim that the method is competitive to be unsupported by the empirical evidence. Other existing methods that are not compared to in this paper offer guaranteed acyclicity and unlimited sampling.

**Resubmission Of Major Revision:**

The authors may consider submitting a major revision at a later time.